# High Prevalence of a Newly Discovered Wutai Mosquito Phasivirus in Mosquitoes from Rio de Janeiro, Brazil

**DOI:** 10.3390/insects10050135

**Published:** 2019-05-07

**Authors:** Mário Sérgio Ribeiro, Tania Ayllón, Viviana Malirat, Daniel Cardoso Portela Câmara, Cristina Maria Giordano Dias, Guilherme Louzada, Davis Fernandes-Ferreira, Roberto de Andrade Medronho, Renata Campos Acevedo

**Affiliations:** 1Institute of Microbiology Paulo de Góes, Federal University of Rio de Janeiro, Av. Carlos Chagas Filho, 373, Rio de Janeiro 21941-970, Brazil; mario_sesrj@yahoo.com.br (M.S.R.); guilouzada@micro.ufrj.br (G.L.); dfferrei@ncsu.edu (D.F.-F.); renatacampos@micro.ufrj.br (R.C.A.); 2Institute of Collective Health Studies, School of Medicine, Federal University of Rio de Janeiro, Av. Horacio Macedo, S/N, Rio de Janeiro 21941-598, Brazil; robertoamedronho@gmail.com; 3Acute Febrile Diseases Laboratory, Evandro Chagas National Institute of Infectious Diseases, Fiocruz, Av. Brasil 4365, Rio de Janeiro 21040-360, Brazil; 4Núcleo Operacional Sentinela de Mosquitos Vetores, Fiocruz, Av. Brasil 4365, Rio de Janeiro 21040-360, Brazil; dcpchamber@gmail.com; 5Animal Virology Center, CONICET-SENASA, Saladillo 2468, Buenos Aires 1440, Argentina; vmaliratcevan@centromilstein.org.ar; 6Laboratório de Mosquitos Transmissores de Hematozoários, Fiocruz, Av. Brasil 4365, Rio de Janeiro 21040-360, Brazil; 7Epidemiological and Environmental Surveillance, State Health Secretariat of Rio de Janeiro, R. México, 128, Rio de Janeiro 20031-142, Brazil; cristina.giordano@saude.rj.gov.br; 8Department of Molecular and Structural Biochemistry, North Carolina State University, 120 W Broughton Dr, Raleigh, NC 27607, USA

**Keywords:** insect-specific viruses, Bunyavirus, *Phasivirus*, *Wutai* virus, entomological surveillance

## Abstract

Many RNA viruses have recently emerged, threatening humans and causing harm to animals and plants. Bunyaviruses represent one of the largest groups of RNA viruses and are able to infect a wide range of hosts (invertebrates, vertebrates, and plants). Recently, new insect-specific viruses have been isolated from mosquitoes and phlebotomine sandflies worldwide. Little is known regarding the impact of these viruses on the vector life cycles and the stages of oviposition, breeding, blood feeding, and the mosquito’s lifespan. This study describes, for the first time in South America, the detection and characterization of a recently discovered bunyavirus corresponding to the Wutai mosquito phasivirus, confirming its high prevalence in the *Culex* spp. and *Aedes* spp. mosquitoes collected in the urban environment of Rio de Janeiro city, Brazil. The knowledge of the mosquito’s insect-specific virus infection can improve virus evolution studies and may contribute to the understanding of intrinsic factors that influence vector competence to transmit pathogenic viruses.

## 1. Introduction

Many RNA viruses have recently emerged, threatening humans and causing harm to animals and plants. Those viruses present a unique capacity to evolve, probably influenced by their large population size, lack of proofreading activity of RNA polymerases, recombination, and reassortment. Moreover, the ability to replicate and be transmitted by arthropods can play a specific role in virus diversity and evolution [1]. 

Bunyaviruses represent one of the largest groups of RNA viruses and are able to infect a wide range of hosts (invertebrates, vertebrates, and plants), which brings an enormous complexity to these viruses. The prototype species (*Bunyamwera orthobunyavirus*) was first isolated from *Aedes* mosquitoes in the Semliki Forest, Uganda, during a yellow fever study in 1943 [2]. This detection was followed by the isolation of several other species, leading to the establishment of the *Bunyaviridae* family in 1975 and the proposal of five genera (*Orthobunyavirus*, *Phlebovirus*, *Nairovirus*, *Hantavirus*, and *Tospovirus*). Except for the hantaviruses, that present rodents as the main reservoir in nature and aerosols as the transmission route, all other genera are transmitted by and replicate in arthropod vectors. Due to the wide range of viruses described and ungrouped in the *Bunyaviridae* family, the International Committee on Taxonomy of Viruses (ICTV) has recently created the order *Bunyavirales* grouping nine families (*Feraviridae*, *Fimoviridae*, *Hantaviridae*, *Jonviridae*, *Nairoviridae*, *Peribunyaviridae*, *Phasmaviridae*, *Phenuiviridae*, and *Tospoviridae*) composed of 13 genera (*Orthoferavirus, Emaravirus, Orthohantavirus, Orthojonvirus, Orthonairovirus, Herbevirus, Orthobunyavirus, Orthophasmavirus, Goukovirus, Phasivirus, Phlebovirus, Tenuivirus*, and *Orthotospovirus*) [3]. 

The bunyaviruses are spherical particles with a negative-sense or ambi-sense single-stranded genome organized into three segments (S, M, and L segments). The S segment codes for the nucleocapsid protein (N); the M segment codes for the two glycoproteins (Gn and Gc), and the L segment codes for the RNA dependent RNA polymerase (RdRp).

Many arthropod-borne bunyaviruses are human and animal pathogens and symptoms can range from a self-limiting disease to a severe hemorrhagic fever. Arthropod-borne agents producing hemorrhagic fevers are mainly members of the *Orthonairovirus* genus (*Nairoviridae* family) and *Phlebovirus* genus (*Phenuiviridae* family), including Crimean-Congo hemorrhagic fever virus (CCHFV), Rift Valley fever virus (RVFV), and Severe Fever with thrombocytopenia syndrome virus (SFTSV). Viruses related to mild febrile illness are mainly members of the *Orthobunyavirus* genus (*Peribunyaviridae* family) [4], many of them detected in the Amazon region of Brazil [5,6,7,8,9]. However, despite the favorable environment for bunyavirus circulation in Brazil, other genera and families in the *Bunyavirales* order remain poorly studied. Recently, new insect-specific viruses have been isolated from mosquitoes and phlebotomine sandflies worldwide, adding information for the proposal of new genera, such as *Goukovirus* [10] and *Phasivirus* [11,12]. 

In this study, we describe, for the first time, the detection and characterization of the Wutai mosquito phasivirus in *Culex* spp. and *Aedes* spp. collected in the urban environment of Rio de Janeiro city, Brazil. This virus was first detected in China in 2012 [1], and only few reports of virus detection and genome sequencing are currently available. Besides, to the authors´ knowledge, this virus has not been isolated yet. The knowledge of insect-specific virus infection can improve virus evolution studies and contributes to the understanding the intrinsic factors that influence vector competence to transmit pathogenic viruses. 

## 2. Materials and Methods

### 2.1. Sample Collection

A prospective study of arbovirus surveillance in mosquitoes was performed from 2013 to 2017 in Rio de Janeiro (collection license: SISBIO 32476-1). To monitor arbovirus presence, 11 health units and seven different places (three residence buildings, three sport complexes, and one public building) were chosen for setting the passive traps, according to the proximity to the Olympic game’s facilities in Rio de Janeiro in 2016. Mosquito trapping was also conducted in two municipalities in the Northern region of the State, Itaperuna and Campos dos Goytacazes, where a high incidence of dengue fever was observed in 2015. The collection was performed weekly using BG-Sentinel traps running for 48 h. The trapped mosquitoes were cooled and carried to the laboratory, counted, sexed, and identified to genus level using the taxonomic key proposed by Consoli and Lourenço-de-Oliveira [13]. Taxonomic confirmation was performed by molecular analyses to confirm the mosquito species [13,14]. Pools up to 250 individuals were prepared for molecular analysis, according to genus, sex, trapping localization, and date. Engorged females were analyzed individually. Moreover, the feed source was determined from samples positive to Wutai mosquito phasivirus [14]. Mosquitoes were homogenized in 1000 µL or 3000 µL of Dulbecco’s Modified Eagle’s Medium (DMEM) (for pools containing 1–25 or 26–250 individuals, respectively) supplemented with 3% of fetal calf serum, 2.5 µg/ml amphotericin B, 500 U/mL penicillin, and 100 µg/mL streptomycin. 

### 2.2. Metagenomics Analysis

Briefly, the mosquito homogenate supernatant was filtered through a 0.45 µm filter (Millipore, Darmstadt, Germany) to remove larger cell debris and bacteria and treated with a mixture of nucleases (Turbo DNase, Ambion, Carlsbad, CA, USA; Baseline-ZERO, Epicenter, Madison, WI, USA; Benzonase, Novagen, San Diego, CA, USA; RNAse One, Promega, Fitchburg, WI, USA) for 1.5 h to digest unprotected nucleic acids including the host DNA/RNA. Enriched viral particles were then extracted, reverse transcribed using random hexamers into double-stranded cDNA, fragmented, ends repaired, dA-tailed, adaptor ligated, and purified. Library preparation was performed using NEBNext^®^ Ultra™ DNA Library Prep Kit for Illumina^®^ (New England Biolabs, Inc. Ipswich, MA, USA). Sequencing was performed using the Illumina Miseq platform. The acquired reads were trimmed, and de-novo assembled using Geneious v9.1.8 (Biomatters, Auckland, New Zealand). The reads and contigs that generated > 100 bp were subjected to homology search using BLASTX against a non-redundant database (http://www.ncbi.nlm.nih.gov/Genbank).

### 2.3. Wutai Mosquito Phasivirus RNA Detection by RT-PCR

One set of primers targeting the L segment was designed based on the complete sequence obtained from the metagenomic study (BunyaBr1- F-383 5′ CTA GAC AAG AGG AAC TAA GTG C 3′; BunyaBr1- R-383 5′ TGT GGG TGC TAG AGA GTG AT 3′). This primer set was used to estimate the Wutai mosquito phasivirus prevalence. The commercial kit QIAamp viral RNA mini (QIAGEN, Valencia, CA, USA) was used to extract RNA according to the manufacturer’s instructions, from 140 µL of the mosquito homogenates. Standard RT-PCR was performed using the Superscript III one-step RT-PCR kit (Invitrogen, Carlsbad, CA, USA) according to the manufacturer´s instructions, with 0.7 µM primers and temperature conditions as follows—60 °C for 1 min, 50 °C for 45 min, and 94 °C for 2 min, followed by 45 cycles of 95 °C for 15 sec, 55 °C for 30 sec, and 68 °C for 45 sec with a final extension of 68 °C for 7 min. 

### 2.4. Nucleotide Sequence Determination and Phylogenetic Analysis

Amplicons were sequenced in the ABI 3730 genetic analyzer (Applied Biosystems, Foster City, CA, USA) following the manufacturer’s protocol. Raw sequence data were aligned, edited, and assembled using the BioEdit Sequence Alignment Editor, Version 7.0.5.3. The phylogenetic trees were based on nucleotide sequences from the L, M, or S segments. Different evolutionary models were tested using the Akaike Information Criteria (AIC) and a Likelihood Ratio Test (LRT) by the means of the program MEGA, to identify the optimal evolutionary model to be applied in each analysis. Maximum likelihood was used with 1000 replicates for bootstrap support of the tree groupings.

## 3. Results

### 3.1. Complete Genome of Wutai Mosquito Phasivirus Detected in Rio de Janeiro

The contigs generated by the metagenomic analysis revealed the presence of bunyavirus genomic material related to a recently described member of the *Phasivirus* genus within the *Phenuiviridae family*, in a group of four *Culex* spp. female. Nucleotide sequences corresponding to the three L, M, and S segments could be identified. The L segment encoding the most conserved viral polyprotein, the polymerase (RdRp), was used to confirm the identity of the detected virus. According to the latest revision of the bunyavirus taxonomy by the ICTV, the demarcation of the species is performed by comparing a fragment of approximately 1 KB in the central domain (premotif A to motif E) within the third conserved region of L gene, considering a new species when less than 90% homology is observed at the amino acid level [3,11]. 

As depicted in Figure 1 the comparative analysis of the deduced amino acid sequence of this fragment (nucleotides 3001 to 4014 of the Wutai mosquito phasivirus virus isolated in Rio de Janeiro) showed 98.1% homology (90.3% at the nucleotide level) with the Wutai mosquito phasivirus detected in China in 2012 from *Culex quinquefasciatus* (GenBank accession number KM 817700) (https://www.ncbi.nlm.nih.gov/pubmed/25633976), and was assigned as the prototype species. Additionally, the Brazilian Wutai mosquito phasivirus presented 96.3% amino acid homology with CBunV1/Kern isolate, another representative of the Wutai mosquito phasivirus species, collected in California, USA, in 2016 from *Culex quinquefasciatus* (GenBank accession number MH188051) (https://www.ncbi.nlm.nih.gov/pubmed/30098450).

An estimation of genetic relationship assessed for the complete coding region of all genomic segments (L, M and S) with the corresponding ones of members of all the four genera in the *Phenuiviridae* family and other bunyavirus families, confirmed the genetic proximity between the Wutai mosquito phasivirus isolates detected in Brazil and China (Figure 2). All nucleotide sequences obtained were deposited at GenBank, reference numbers are shown at the sequence title. 

### 3.2. Wutai Mosquito Phasisvirus Prevalence in Rio de Janeiro and Phylogenetic Analysis

A total of 1866 adult mosquitoes trapped between 2013 and 2017 were randomly selected from the laboratory mosquito surveillance study for bunyavirus screening. Of these, 1465 (78.51%) were identified as the *Culex* spp. (824 males, 418 non-engorged females, and 223 engorged females) and 401 (21.49%) were identified as the *Aedes* spp. (150 males, 248 non-engorged females, and 3 engorged females). The geographical distribution of sampling locations is presented in Figure 3. 

The Wutai mosquito phasivirus was detected in 42 of 149 pools analyzed (Table 1 and Table 2, Appendix A). The virus was detected in male mosquitoes and in non-engorged and engorged females. A total prevalence of 4.03% was observed in *Culex* engorged females (Table 1). 

The identification of the blood meal sources from engorged positive females revealed *Homo sapiens* and *Canis lupus familiaris* as the feeding sources. The molecular analyses confirmed the mosquito species contained in the positive pools. Complex *Culex pipens quinquefasciatus* were confirmed positive for the Wutai mosquito phasivirus. Another positive pool was comprised of the *Culex mollis* species (genus *Culex*, subgenus *Culex*) and *Conspirator* group (genus *Culex*, subgenus *Melanocolion*), pointing out these groups as possible hosts for the Wutai mosquito phasivirus. Only one pool of male *Aedes aegypti* also presented positive results. The complete genome referred to previously (Figure 1 and Figure 2) was obtained from one out of the 42 positive pools, whereas the partial sequence from the L segment was determined for an additional 13 positive pools. All the partial sequences obtained in this work clustered together and presented a homology ranging from 97.7 to 99.7% (Figure 4).

## 4. Discussion

This study reports, for the first time, the complete genome of a Wutai mosquito phasivirus detected in South America. This species belongs to a recently described genus in which four species were included (Wutai mosquito phasivirus, Badu Virus, Wuhan fly, and Phasi Charoen-like) with only few complete genomes available in the Genbank. 

Our results also demonstrated that the virus can infect different culicids. At least three species from the *Culex* genus were positive (*Culex mollis, Culex conspirator, and Culex pipens quinquefaciatus*). A pool containing *Ae aegypti* males was also positive, strengthening the wide distribution of this insect-specific virus among mosquito species. There is no information regarding the route of transmission of the Wutai mosquito phasivirus and the impact on the ecology of the mosquito hosts. All the strains detected in Rio de Janeiro clustered together, independently of the host; however, further studies targeting variable regions of the genome are needed in order to evaluate the diversity and evolution of this virus. 

In 2015, one study revealed the presence of *Phasivirus* in Brazil. The sequencing, based on the pattern of small viral RNAs, revealed the presence of a *Phasi Charoen-like virus* (PCLV) in *Aedes aegypti* individuals obtained from colonies established from mosquitoes trapped in Rio de Janeiro city [12], a virus originally identified in mosquitoes from Thailand. One year later, the same virus was isolated from *Ae. aegypti* mosquitoes in northern Australia [17]. 

Beyond the diversity of mosquito species infected, the Wutai mosquito phasivirus presented a wide geographic distribution in Rio de Janeiro. From a total of 13 locations screened in this study, nine locations had mosquitoes positive for this virus. Nonetheless, the four locations that tested negative for *Phasivirus* (4, 5, 8, and 13) cannot be considered negative due to the low number of mosquitoes analyzed *per* area.

Questions remain regarding the impact of the Wutai mosquito phasivirus in the vector life cycle and the stages of oviposition, breeding, blood feeding, and mosquito’s lifespan. Recently, the studies of vector competence have been highlighting the influence of microbial communities on the ability of mosquito populations to transmit arboviruses [18]. A well-characterized microbial-arboviral relationship is an inhibition promoted by *Wolbachia* on the transmission of dengue viruses. Aside from bacteria and fungi, the influence of the insect virome on vector competence should be better investigated. Efforts should be done in order to isolate the insect-specific viruses described herein to enable experiments to evaluate the influence on vector competence. 

Bunyaviruses are an important group for the emergence of pathogenic strains, many of them transmitted by blood-feeding vectors. Therefore, efforts should be done to improve the knowledge of bunyavirus genomic diversity, including insect-specific strains, in order to complete viral evolution studies and add clues to understanding the origin of pathogenic strains. 

## 5. Conclusions

This study describes for the first time in Brazil the detection and characterization of a recently discovered bunyavirus, the Wutai mosquito phasivirus. The study also confirms the presence of this virus in the main urban vectors for the arboviruses, *Culex* spp. and *Aedes* spp. A high prevalence and a wide distribution of infected *Culex* spp. were observed in Rio de Janeiro city, contributing to a better understanding of the mosquito’s insect-specific virus distribution. This information also contributes to virus evolution studies and should be considered in studies that determine vector competence to transmit pathogenic viruses. 

## Figures and Tables

**Figure 1 insects-10-00135-f001:**
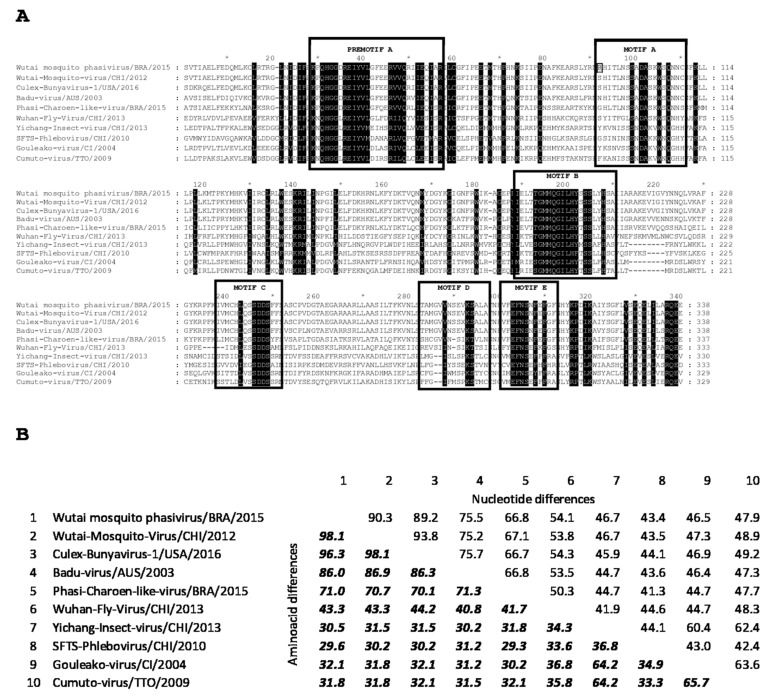
Sequence alignment of RdRp third conserved motif of selected viruses. (**A**) Amino acid alignment showing pre-motif A and motifs A, B, C, D, and E, highlighted by boxes. Highly conserved amino acid residues between Wutai-Mosquito-Phasivirus/BRA/2015 and selected bunyaviruses are marked in black. The only difference found within the motifs between Wutai-Mosquito-Phasivirus/BRA/2015 and the species prototype strain Wutai-Mosquito-Virus/CHI/2012, in position 92 of the alignment is highlighted. Sequence alignment position is indicated in an upper marker line, every 10 residues. Number 1 in the alignment corresponds to amino acid position 1001 of the Wutai-Mosquito-Phasivirus/BRA/2015 and Badu-Virus/AUS/2003; 949 of Wutai-Mosquito-Virus/CHI/ and Culex-Bunyavirus-1/USA/2016; 1002 of Phasi-Charoen-like-Virus/BRA/2015; 970 of Wuhan-Fly-Virus/CHI/2013; 909 of Yichang-Insect-Virus/CHI/2013; 885 of SFTS-Phlebovirus/CHI/2010; 880 of Gouleako-Virus/CI/2004; and 895 of Cumuto-Virus/TTO/2009. Numbers at the end of each line indicates the total number of amino acids in the corresponding row. (**B**) Pairwise comparison of percentage of differences per site, calculated at the nucleotide (upper right) and amino acid (lower left) levels between the Wutai-Mosquito-Phasivirus/BRA/2015 and the selected bunyaviruses are shown. All the positions containing gaps and missing data were eliminated. There were a total of 966 nucleotide and 321 amino acid positions in the final datasets.

**Figure 2 insects-10-00135-f002:**
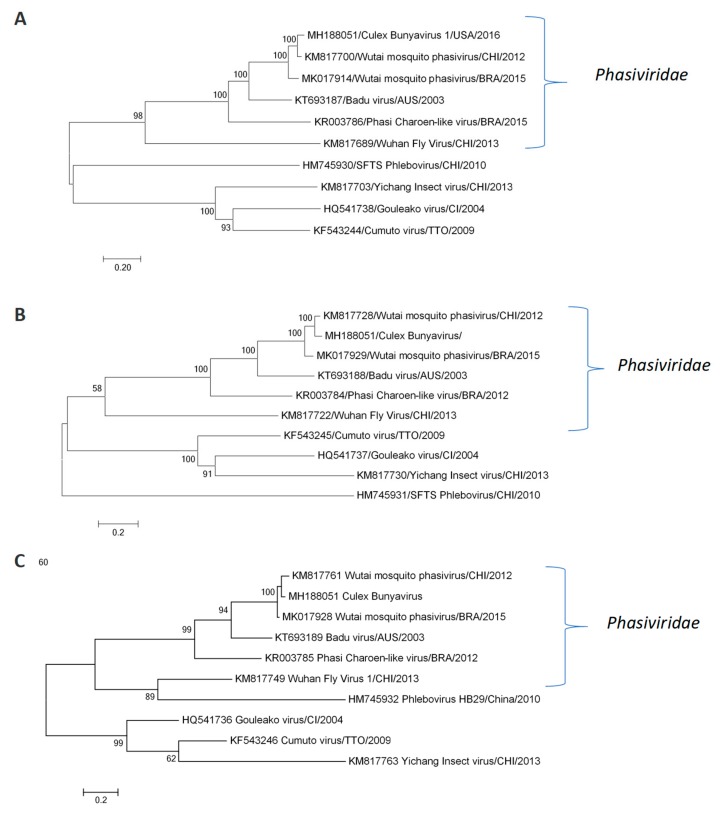
Genetic relationship of Wutai mosquito phasivirus with other members of the *Phasiviridae* family and other related bunyaviruses. Phylogenetic trees were constructed for (**A**) L segment (5800 bp); (**B**) M segment (2620 bp); and (**C**) S segment (511 pb). The evolutionary history was inferred by using the Maximum Likelihood method based on the General Time Reversible model for segments L and M and on the Tamura 3-parameter model for segment S [15]. A discrete Gamma distribution was used to model evolutionary rate differences among sites (5 categories). The rate variation model allowed for some sites to be evolutionarily invariable for segments L and M. The trees are drawn to scale, with branch lengths measured in the number of substitutions per site. All positions containing gaps and missing data were eliminated. Evolutionary analyses were conducted in MEGA7 [16], with 1000 replicates. Only bootstrap values > 70% are shown at the node.

**Figure 3 insects-10-00135-f003:**
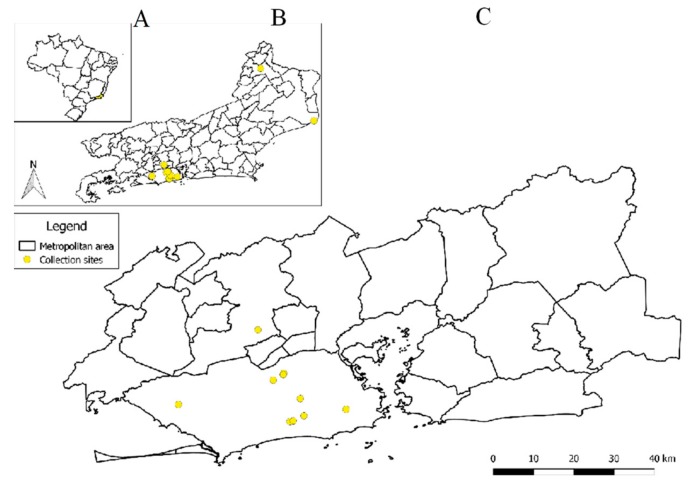
Geographic distribution of trapping locations. (**A**) Brazil; (**B**) Rio de Janeiro state; (**C**) Metropolitan region.

**Figure 4 insects-10-00135-f004:**
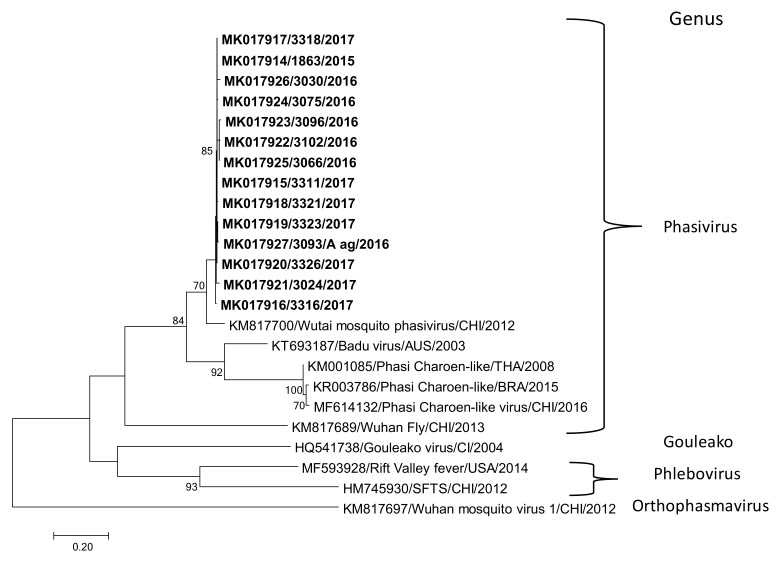
Molecular phylogenetic analysis of Wutai mosquito phasivirus. The phylogenetic tree was constructed by using partial nucleotide sequence of the L segment (315 bp). The evolutionary history was inferred by using the Maximum Likelihood method based on the Tamura 3-parameter model. A discrete Gamma distribution was used to model the evolutionary rate differences among the sites (five categories (+*G*, parameter = 37,057)). The tree is drawn to scale, with branch lengths measured in the number of substitutions per site. Evolutionary analyses were conducted in MEGA7, with 1000 replicates. Only bootstrap values > 70% are shown at the node. The bold-sequences were determined in this study.

**Table 1 insects-10-00135-t001:** Distribution of the Wutai mosquito phasivirus positive (pos) *Culex* spp. pools according to the trapping locations.

Locations	Males *n*	Pools *n* (pos)	Non-Engorged Females *n*	Pools *n* (pos)	Engorged Females	
					*n*	Pos *n* (%)
1	107	12 (6)	74	11 (8)	0	-
2	180	4 (2)	77	4 (1)	47	1 (2.1%)
3	21	1 (1)	55	4 (0)	21	0
4	0	-	0	-	2	0
5	0	-	15	1 (0)	21	0
6	49	4 (2)	43	4 (3)	0	-
7	5	1 (1)	12	2 (1)	6	0
8	0	-	0	-	3	0
9	32	3 (0)	18	2 (1)	31	0
10	347	2 (2)	98	1 (1)	15	0
11	0	-	0	-	74	8 (10.81%)
12	83	4 (3)	26	2(0)	0	-
13	0	-	0	-	3	0
**Total**	824	31 (17)	418	32 (15)	223	9 (4.03%)

**Table 2 insects-10-00135-t002:** Distribution of the Wutai mosquito phasivirus positive (pos) *Aedes* spp. pools according to the trapping locations.

Locations	Males *n*	Pools *n* (pos)	Non-Engorged Females *n*	Pools *n* (pos)	Engorged Females	
					*n*	Pos *n* (%)
2	64	13 (0)	22	11 (0)	0	-
3	33	10 (0)	20	7 (0)	0	-
4	0	-	5	3 (0)	0	-
7	13	7 (0)	16	7 (0)	0	-
9	26	9 (0)	28	11 (0)	1	0
10	14	4 (1)	157	4 (0)	0	-
13	0	0	0	0	2	0
Total	150	43 (1)	248	43 (0)	3	0

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
