# Peer review of "High Prevalence of a Newly Discovered Wutai Mosquito Phasivirus in Mosquitoes from Rio de Janeiro, Brazil"

_insects, 2019, doi:10.3390/insects10050135_

Round 1

Reviewer 1 Report

In the article ‘High Prevalence of a Newly Discovered Wutai Mosquito Phasivirus in Mosquitoes from Rio de Janeiro, Brazil’ by Ribeiro et al. submitted to Insects, the authors describe the identification of a recently discovered bunyavirus in Brazilian mosquitoes. There is nothing fundamentally wrong with the study design and execution, but the knowledge gained is incremental.

Recently, more and more of these types of papers are being published ‘we found a virus in our mosquitoes here’. I believe this is not unimportant and the study is probably suitable for publication in ‘Insects’, however, I urge the authors to characterize these viruses further in the lab in future studies. At this point it is not much more than a genome announcement. The manuscript is overall relatively well-written but could use some improvements as described below.

Specific comments:

Figure 1B is unclear and the description most likely wrong. The figure legend says that the ‘lower left’ is nucleotide comparison, and the ‘upper right’ amino acid. Based on the numbers this is exactly mixed up. 90.3 (upper right comparison between 1 and 2) would be the nucleotide level relation. Please label this figure more clearly so that it is intuitive without the legend. And why are the viruses not compared to ‘10’ Cumuto-virus on the nucleotide level (columns)?

In line 75-77, the authors introduce Wutai mosquito phasivirus, yet do not give a citation of its first discovery. The authors later refer to the original isolate from China, so there should be a citation here and it would be useful if the authors also mentioned in a sentence or two where was previously found. (and was it isolated or just sequences identified?). All of this would be useful information provided right here in the introduction.

Line 95: what are the ‘positive samples’? positive for a blood meal? Positive for any arboviruses? Positive for dengue? I suggest to rephrase and be clear.

Line 101: ‘mixture of nucleases’ please be specific and say which nucleases were used and how long samples were incubated for DNase/RNase treatment. This information is important for the reader to correctly interpret the data.

Please stay consistent with the use of ‘Wutai mosquito phasivirus’ throughout – the authors often use ‘Wutai mosquito virus’ or ‘Wutai virus’ interchangeably with ‘Wutai mosquito phasivirus’. These names also do not need to be italicized here. Also the authors essentially say that the Chinese isolate is the same species due to the high amino acid similarity. Yet they use different names for these viruses in their phylogenetic tree. Please keep all of this consistent throughout.

Lines 143-146: There publications for these viruses. These should be cited along with the Genbank numbers (https://www.ncbi.nlm.nih.gov/pubmed/25633976 and https://www.ncbi.nlm.nih.gov/pubmed/30098450).

Some minor stylistic English improvements could be beneficial throughout the manuscript. Examples and/or suggestions are below:

Lines 27-28 (as well as 45-47): ‘Bunyaviruses represent one of the largest groups of RNA viruses and are able to infect …

Line 35: ‘… and may contribute to the understanding of intrinsic factors that…’

Line 36/79: vector competence instead of vector’s competence (or ‘a vector’s competence …’)

Line 63: ‘…and symptoms of bunyavirus infections can range from a self-limiting disease to severe hemorrhagic fever’. (or similar)

Line 73: ‘…adding information for the proposal of new genera’.

Line 83: ‘arbovirus surveillance’ instead of ‘arboviruses surveillance’.

Line 108: should be 100bp instead of 100pb.

Line 149: typo, should be ‘Amino acid’ (in line 150 amino acid is also spelled as one word)

Line 173: should be ‘other related bunyaviruses’

Line 250: I suggest to replace the word ‘powerful’ with something less ‘dramatic’ – maybe important or similar…

Author Response

ANSWERS TO REVIEWER # 1

The authors thank reviewer #1 for the exhaustive and positive evaluation of the manuscript and for the suggestions to improve understanding of the paper.

Point 1: Figure 1B is unclear and the description most likely wrong. The figure legend says that the ‘lower left’ is nucleotide comparison, and the ‘upper right’ amino acid. Based on the numbers this is exactly mixed up. 90.3 (upper right comparison between 1 and 2) would be the nucleotide level relation. Please label this figure more clearly so that it is intuitive without the legend. And why are the viruses not compared to ‘10’ Cumuto-virus on the nucleotide level (columns)?

Response 1: We have corrected the errors pointed out by the reviewer #1.

1.     Interchange of “upper right” and “lower left” when referring to the pairwise distances.

2.     Inclusion of the column with the ’10’ Cumuto-virus nucleotide differences, which was missing in the original subscription.

Besides, we have labeled the figure more clearly to make it more intuitive, as suggested.

Point 2: In line 75-77, the authors introduce Wutai mosquito phasivirus, yet do not give a citation of its first discovery. The authors later refer to the original isolate from China, so there should be a citation here and it would be useful if the authors also mentioned in a sentence or two where was previously found. (and was it isolated or just sequences identified?). All of this would be useful information provided right here in the introduction.

Response 2: This information was added in the introduction section.

Point 3: Line 95: what are the ‘positive samples’? positive for a blood meal? Positive for any arboviruses? Positive for dengue? I suggest to rephrase and be clear.

Response 3: This refers to Wutai mosquito phasivirus positive samples (current lines: 118-119). This sentence was rephrased in the text for better clarity.

Point 4: Line 101: ‘mixture of nucleases’ please be specific and say which nucleases were used and how long samples were incubated for DNase/RNase treatment. This information is important for the reader to correctly interpret the data.

Response 4: We have included this information in the text.

Point 5: Please stay consistent with the use of ‘Wutai mosquito phasivirus’ throughout – the authors often use ‘Wutai mosquito virus’ or ‘Wutai virus’ interchangeably with ‘Wutai mosquito phasivirus’. These names also do not need to be italicized here. Also the authors essentially say that the Chinese isolate is the same species due to the high amino acid similarity. Yet they use different names for these viruses in their phylogenetic tree. Please keep all of this consistent throughout.

Response 5: All Wutai mosquito phasivirus citation was revised. The name was corrected according to ICTV guidelines. https://talk.ictvonline.org/information/w/faq/386/how-to-write-a-virus-name

Point 6: Lines 143-146: There publications for these viruses. These should be cited along with the Genbank numbers (https://www.ncbi.nlm.nih.gov/pubmed/25633976 and https://www.ncbi.nlm.nih.gov/pubmed/30098450).

Response 6: As suggested, the publications were cited in the text after Genbank numbers.

Point 7: Some minor stylistic English improvements could be beneficial throughout the manuscript. Examples and/or suggestions are below:

Lines 27-28 (as well as 45-47): ‘Bunyaviruses represent one of the largest groups of RNA viruses and are able to infect …

Line 35: ‘… and may contribute to the understanding of intrinsic factors that…’

Line 36/79: vector competence instead of vector’s competence (or ‘a vector’s competence …’)

Line 63: ‘…and symptoms of bunyavirus infections can range from a self-limiting disease to severe hemorrhagic fever’. (or similar)

Line 73: ‘…adding information for the proposal of new genera’.

Line 83: ‘arbovirus surveillance’ instead of ‘arboviruses surveillance’.

Line 108: should be 100bp instead of 100pb.

Line 149: typo, should be ‘Amino acid’ (in line 150 amino acid is also spelled as one word)

Line 173: should be ‘other related bunyaviruses’

Line 250: I suggest to replace the word ‘powerful’ with something less ‘dramatic’ – maybe important or similar…

Response 7: The authors thank the reviewer for the stylistic English suggestions and have modified all the sentences accordingly.

Reviewer 2 Report

High Prevalence of a Newly Discovered Wutai Mosquito Phasivirus in Mosquitoes from Rio de Janeiro, Brazil

Ribeiro et al describe the discovery of Wutai Mosquito Phasivirus in mosquitoes from Rio de Janeiro. In addition, the paper also confirms the presence of this newly discovered virus in Culex and Aedes mosquitoes.

The paper is interesting and well written. The experiments are well done and presented. The data throws light into the largest but one of the least studied family of RNA viruses. It also provides solid bases for further research, such as the application of mosquito specific bunyaviruses for vector or virus control. In addition, it seems that these viruses do not co-evolve with the mosquito hosts but that they are transmitted among species.

Overall I enjoyed the paper, the data is sound and the findings provide a good base for future studies with this novel viruses.

Author Response

ANSWER TO REVIEWER # 2

We thank reviewer #2 for the positive evaluation of the manuscript.
